# Transcriptomic Study of Spermatogenesis in the Testis of Hu Sheep and Tibetan Sheep

**DOI:** 10.3390/genes13122212

**Published:** 2022-11-25

**Authors:** Xiaoyu Fu, Yanan Yang, Zunqiang Yan, Miaomiao Liu, Xinrong Wang

**Affiliations:** College of Animal Science and Technology, Gansu Agricultural University, Lanzhou 730070, China

**Keywords:** Hu sheep, Tibetan sheep, testis tissue, histological observation, RNA-Seq

## Abstract

Numerous genes involved in male reproduction regulate testis development and spermatogenesis. In this study, the testis tissue transcriptome was used to identify candidate genes and key pathways associated with fecundity in sheep. Histological analysis of testis tissue using hematoxylin–eosin (HE) routine staining was performed for two sheep breeds. Overall, 466 differentially expressed genes (DEGs) were identified between Hu sheep (HS) and Tibetan sheep (TS) through RNA sequencing technology (RNA-Seq), including 226 upregulated and 240 downregulated genes. Functional analysis showed that several terms and pathways, such as “protein digestion and absorption”, “cAMP signaling pathway”, “focal adhesion”, and “p53 signaling pathway” were closely related to testis development and spermatogenesis. Several genes (including *COL1A1*, *COL1A2*, *COL3A1, SOX9*, *BCL2*, *HDC*, and *GGT5*) were significantly enriched in these terms and pathways and might affect the reproduction of sheep by regulating the migration of spermatogenic cells, apoptosis of spermatogenic cells, and secretion of sterol hormones via testicular interstitial cells. Our results provide a theoretical basis for better understanding the molecular mechanisms of reproduction in sheep.

## 1. Introduction

A major limitation to the development of animal husbandry is the reproductive capacity of sheep. However, most sheep are singleton, seasonal breeds [1]. A sheep’s fertility is determined by the reproductive contribution of both rams and ewes. When the ewes are identical, ram fertility is especially important. The testis is a highly specialized tissue in male mammals that ensures fertility by producing sperm and androgenic hormones; both processes are influenced by many factors and fundamentally regulated by the expression of a large number of genes encoding proteins, which are developmentally regulated during spermatogenesis and play key roles in specific stages of germ cell development [2]. Therefore, it is critical to identify the regulatory genes affecting testis development and spermatogenesis in sheep.

Tibetan sheep are one of the three original sheep breeds in China and are mainly distributed in the Qinghai–Tibet Plateau and its surrounding areas [3]. Tibetan sheep live in a high-altitude hypoxic environment. Their reproductive physiology is characterized by low fertility and late sexual maturity [4]. Hu sheep are a famous multiparous breed that mainly lives in the eastern coastal provinces with low altitudes in China. They have large body sizes, are highly fertile, and have strong adaptability [5]. Although the disparity between the reproductive rates of these breeds might be due to differences in the environment or foraging or breeding methods, genetic discrepancies may also be a major factor. Hence, understanding the mechanisms underlying the reproductive differences between breeds is vital for the sheep industry.

High-throughput RNA sequencing technology (RNA-Seq) is an effective technique that can be used to study gene function and structure at the holistic level [6]. A variety of animals have been studied using this technique to explore reproductive mechanisms, including humans [7], drosophilae [8], mice [9], goats [10], and boars [11]. For instance, 24,878 differentially expressed genes (DEGs) were detected in the immature and mature testes of the Mongolian horse, and five genes with alternative splicing events that may influence spermatogenesis and the development of the testis were detected [2]. However, studies on the differences in the expression of sheep testes’ monomers and polypeptides have not been reported and thus require investigation.

The testicular tissue of Tibetan sheep and Hu sheep were used in the current study as experimental samples. RNA-Seq and bioinformatic analyses were conducted to explore the molecular mechanisms of different fecundities in sheep. Additionally, this study aimed to provide relevant biomarkers and a theoretical basis for improving Tibetan sheep fertility.

## 2. Materials and Methods

### 2.1. Ethics Statement

This study was performed in accordance with the guidelines of the Animal Care and Use Committee of the Biological Research of Gansu Province, China. The Ethics Committee of Gansu Agricultural University approved all the animal procedures (Approval number 2019-044). The animals did not suffer unnecessarily.

### 2.2. Animals and Experimental Design

A total of 12 male Tibetan sheep (TS) and 12 male Hu sheep (HS) were raised at Linxia (Gansu, China) and Minqin (Gansu, China), respectively. All lambs were immunized using standardized procedures before weaning at 56 days of age and were raised in a separate enclosure of 0.8 × 1 m until they were 1 year old. All experimental individuals were under the same feeding conditions. Three 1-year-old male Hu sheep and three 1-year-old male Tibetan sheep were randomly selected from the 12 HS and 12 TS. Information about the experimental animals is shown in Table 1. The scrotal circumference of all lambs was measured and recorded in the morning. After slaughter, the morphological characteristics of individual testes (including weight, major axis, and minor axis) were measured and recorded using a soft ruler, vernier caliper (0–150 mm, Hengmei, Hangzhou, China), and electronic scale (0.01 g accuracy, Joanlab, Huzhou, China). The right testicular tissue was collected from all sheep for transcriptomic analysis and subsequent validation. Half the tissue samples collected from the testes were fixed in 4% paraformaldehyde for histological analysis, and the other half were immediately frozen in liquid nitrogen and stored at −80 °C for future use in experiments.

### 2.3. Preparation and Data Analysis of Testicular Tissue Sections

The testicular tissue was washed with running water, immersed in wax blocks, cut into 5–7 μm sections, and stained using the hematoxylin and eosin (HE) technique. The sections were observed and photographed using an Olympus DP 71 microscope (Olympus Optical Co., Ltd., Tokyo, Japan). Image Pro-Plus 6.0 software was used to measure the diameter of the seminiferous tubules, the thickness of the seminiferous epithelium, and the area of various cells in the transverse sections of 50 randomly selected seminiferous tubules, and the number of various cells in 50 curved seminiferous tubules was also measured [4]. Mean ± SD (standard deviation) was used to express the measurement data. Student’s *t*-tests were performed using SPSS software (version 25.0) for statistical analysis of the data. A *p*-value < 0.05 was considered statistically significant.

### 2.4. RNA Extraction, cDNA Library Preparation, and RNA-Seq

Total RNA from each sample was extracted using Trizol (TransGen Biotech, Beijing, China) from the testes according to the manufacturer’s instructions. RNA sample integrity and concentration were examined using an Agilent 2100 (Agilent Technologies, Santa Clara, CA, USA). The 260/280 OD values of samples between 1.8 and 2.0 were selected for further analysis. The qualified RNA was used for library construction. A total amount of 1 µg RNA per sample was used for library preparation. The specific procedure was as follows: first, mRNA was isolated using Oligo(dT)-attached magnetic beads and then randomly fragmented in a fragmentation buffer. Fragmented mRNA and random hexamers were used as templates and primers, respectively, to synthesize the first strand of cDNA. Second-strand cDNA was subsequently synthesized using PCR buffer, dNTPs, RNase H, and DNA polymerase I. The products were purified using AMPure XP beads, and double-stranded cDNA was subjected to end repair. To inspect the concentration and insert size of the established library, Qubit 2.0 and Agilent 2100 were used. The constructed cDNA libraries were sequenced using the Illumina sequencing platform.

### 2.5. Gene Expression Quantification and Identification of Differentially Expressed Genes (DEGs)

To obtain high-quality clean data, the raw data were filtered by removing the adapter sequences and low-quality reads. The clean data were further mapped to the ovine reference genome to generate mapped data (ARS-UI_Ramb_v2.0). Gene expression levels were estimated by fragments per kilobase of transcript sequence per million fragments mapped reads (FPKM). All the raw data obtained in this study were deposited in the NCBI Sequence Read Archive (SRA) under the Bioproject accession number PRJNA841692.

DESeq software was used for differential expression analysis of RNA-Seq expression profiles with three biological replications based on a negative binomial distribution. Genes with a *p* value < 0.05 and fold changes (FC) > 1.5 were considered differentially expressed.

### 2.6. Functional Enrichment Analysis of DEGs

The Clusters of Orthologous Groups of Proteins (COG) database was used to classify DEGs. Gene Ontology (GO) terms and Kyoto Encyclopedia of Genes and Genomes (KEGG) pathway analysis were used to investigate the associations between the identified DEGs and gene-related biological functions, and a *p* value < 0.05 was considered significantly enriched. Based on the results of GO and KEGG analyses and biological significance, we selected genes for a follow-up study. Protein–protein interaction (PPI) analysis was performed using STRING-db server [12] (http://string.embl.de/, accessed on 30 May 2022) and visualized using Cytoscape software (version 3.7.1) [13].

### 2.7. RT-qPCR Analysis for the Validation of RNA-Seq Data

Quantitative real-time-PCR (RT-qPCR) was used to detect the mRNA levels of ten genes. The same RNA samples were used for reverse transcription using a reverse transcriptase kit (Takara, Dalian, China) to synthesize first-strand cDNA. Oligo 7.0 and Primer 5.0 were used to design the primers. Sheep β-actin (NM_001009784.3) was used as the internal control (Table 2). The RT-qPCR cycling parameters were as follows: 95 °C for 3 min, followed by 40 cycles of 15 s at 95 °C, the optimized annealing temperature for 15 s, and 72 °C for 20 s. Three biological replicates were used for each assay. The relative expression of the target genes was analyzed using the 2^−∆∆Ct^ method [14].

## 3. Results

### 3.1. Morphologic Parameters of Testis

The morphological indices of the testes showed that the average weight of six testes (including the epididymis) of three Hu sheep was 210.24 g, and the average weight of six testes of three Tibetan sheep was 147.09 g. There was an obvious difference between these values, as well as the measured values of other morphologic indices of the testes (Figure 1).

### 3.2. Histological Observation of Testis

The results of the histological observations of the testes of sheep with different fecundity levels are shown in Figure 2. Compared with Hu sheep, Tibetan sheep have a more compact arrangement of seminiferous tubules, with more Leydig cells and blood vessels between the seminiferous tubules. Spermatogenic, Sertoli, and myoid cells were observed in the seminiferous tubules. However, as is shown in Table 3, the diameter, area, and spermatogenic epithelium thickness of the seminiferous tubules of HS were significantly greater than those of TS (*p* < 0.05). The area of spermatogonia cells and primary spermatocytes of HS was significantly larger than that of TS (*p* < 0.05), and the number of Sertoli cells was higher in Hu sheep (*p* < 0.05).

### 3.3. Summary of the RNA-Seq Data

Six separate cDNA libraries were constructed from the testicular tissue RNA, and the summary statistics of the sequenced RNA-Seq data are shown in Table 4. In total, 266,009,566 raw reads were generated from the six libraries, and 133,004,783 clean reads were left after quality control. The GC contents of the six samples ranged from 49.61–51.14%, which is in accordance with base composition rules. A Q30 ≥ 92.96% indicated that the data were reliable for further analysis. Clean data were aligned to the reference genome, and over 95.97% of the reads were accurately aligned with a high matching rate. About 2.36–2.57% of these clean reads had multiple aligned positions, and 93.45–94.99% of them had single aligned positions. The reads that aligned with the sheep reference genome at a single position were used for further bioinformatic analysis. Through the comparison of the FPKM values of all genes, the overall gene expression levels of the reference genome were used for further analysis. The distribution of gene expression levels in different samples is shown in Figure 3A. A principal component analysis (PCA) of all mapped genes showed that the HS and TS sheep breeds could be distinguished by breed along the axis of the first principal component (Figure 3B).

### 3.4. Functional Categories of the Identified Genes

To further study the function of the DEGs, we classified the identified testicular genes into different categories, as shown in Figure 4. In the COG categories of the identified genes, 5159 of the 5302 genes were classified into groups. Among them, general function prediction only (632 genes); posttranslational modification, protein turnover, and chaperones (591 genes); and signal transduction mechanisms (551 genes) were the most common. Some of these genes were classified as having an unknown function.

### 3.5. Analysis of Differentially Expressed Genes

DEGs between libraries were screened based on DESeq analysis, and transcripts with FC > 1.5 and *p* < 0.05 were classified as DEGs. A total of 466 genes were found to be differentially expressed. Compared with the HS group, 226 genes were upregulated in the TS group and 240 genes were down-regulated (Figure 5A). To understand the differences in the gene expression patterns between sheep in the HS and TS groups, clustering analysis of the DEGs was conducted. The results showed that the DEGs of sheep in the HS and TS groups were divided into two clusters (Figure 5B). Moreover, the level of *BCL2* gene expression had a similar trend to the number of spermatogenic cells and Sertoli cells in both groups (Figure 6).

### 3.6. GO Enrichment Analysis of DEGs

The differentially expressed genes in the HS and TS groups were enriched in 49 GO terms (Appendix A). A total of 20 GO terms belonged to “biological processes”, and these were mainly focused on “cellular process” (173 genes), “single-organism process” (144 genes), and “biological regulation” (120 genes). A total of 16 GO terms belonged to the functional category of “cellular component,” and these were mainly concentrated in the “cell” (171 genes), “cell part” (171 genes), and “membrane” (119 genes) classification groups. In addition, 13 GO terms belonged to “molecular function”, and they centered on “binding” (175 genes), “catalytic activity” (92 genes), and transporter activity (19 genes, Figure 7).

### 3.7. KEGG Pathway Enrichment Analysis of DEGs

Kyoto Encyclopedia of Genes and Genomes (KEGG) pathway enrichment analysis was performed on these DEGs. The results showed that 255 pathways were enriched in the HS vs. TS comparison (Appendix A). Figure 8 shows the results of the top 20 significantly enriched KEGG pathways. The DEGs were mainly enriched in “protein digestion and absorption”, “cAMP signaling pathway”, and “focal adhesion”. Furthermore, we found that some pathways might be involved in testis development and spermatogenesis; they include protein digestion and absorption (*COL1A1*, *COL1A2*), the cAMP signaling pathway (*SOX9*), and the p53 signaling pathway (*BCL2*).

### 3.8. Protein–Protein Interaction Network Analysis

Protein–protein interactions were assessed using the bioinformatics STRING database (https://string-db.org/, accessed on 30 May 2022) and visualized using Cytoscape software based on the 14 DEGs in the HS vs. TS comparison (Figure 9A). In this study, red indicates up-regulated DEGs and blue indicates down-regulated DEGs. The network demonstrated that the *COL1A1* gene interacts with the highest number of genes (eight interactions), followed by *COL3A1* (seven interactions), *COL4A6*, *COL6A2*, *COL4A4*, and *COL5A1* (six interactions).

### 3.9. Validation of RNA-Seq Data by RT-qPCR

To verify the RNA-Seq results, ten DEGs were selected for confirmatory analysis using RT-qPCR (Figure 9B). The results showed that there were some differences in their expression levels, but the expression patterns were consistent, indicating that the RNA-Seq data was reliable.

## 4. Discussion

Fertility plays an important role in animal husbandry. Genetics is an important factor responsible for the differences in reproduction rates. RNA-Seq has emerged as an efficient, high-resolution, and deep-coverage method for detecting transcripts and genes, and it has been shown to be a viable approach in animal studies. With improvements in detection techniques, a large number of genes related to fecundity have been identified. However, analysis of sheep with different reproductive testis transcriptomes has rarely been performed. Tibetan sheep and Hu sheep are two different breeds with different fertility rates. Analysis of the histological structure and gene expression in the sheep testes can provide a comprehensive understanding of the regulatory mechanisms underlying differences in fecundity. Three Hu sheep and three Tibetan sheep were selected for this study, from which key genes and pathways affecting testicular and fertility differences were screened using RNA-Seq. RT-qPCR was performed to verify the transcriptome results.

In the present study, 466 genes were differentially expressed. Functional enrichment results showed that differential genes were significantly enriched in protein digestion and absorption, the cAMP signaling pathway (*SOX9*), focal adhesion (*COL1A1*, *COL1A2*), and the p53 signaling pathway *(BCL2*). Protein digestion and absorption pathways were the most significantly enriched pathways in the KEGG analysis. Studies have shown that protein digestion and absorption are important for testicular metabolite synthesis [15], and genes related to protein metabolism are highly expressed in the testes of mice [16]. Therefore, this pathway may be a major contributor to testis development. Focal adhesion mediates the junction between cells and the extracellular matrix and plays a key role in many cellular functions [17]. Recent studies have shown that the migration of spermatogenic cells in the spermatogenic epithelium is related to the remodeling of anchored junctions between cells during spermatogenesis. This junction is involved in spermatogenesis and sperm release [18]. Collagens belong to a family of extracellular matrix proteins that are essential structural components of tissue and organ development [19,20]. In our analysis, *COL1A1* and *COL1A2* genes were significantly upregulated in the Hu sheep group and enriched in the focal adhesion pathway. The precursor of type I collagen sequences are encoded by two different genes: *COL1A1* and *COL1A2* [21]. During spermatogenesis, *COL1A1* and *COL1A2* regulate the adhesion of spermatogonia and preleptotene spermatocytes to the basement membrane and the detachment and migration of spermatocytes and sperm cells to the lumen at a later stage [22]. In our study, *COL1A1* and *COL1A2* were upregulated in Hu sheep. PPI network regulation revealed that the *COL1A1* gene is located at the central node of the network and interacts with other genes. These results indicate that Focal adhesion mediates a junction between spermatogenic cells and the extracellular matrix, which promotes the migration of spermatogenic cells to the lumen and facilitates spermatogenesis and the release of sperm. Reticulin fibers are fine fibers containing mainly collagen type III. Reticulin fibers in the peritubular tissue of the germinal tubules provide a supportive framework for the germinal epithelium to ensure normal sperm migration to the central lumen of the germinal tubules during spermatogenesis [23]. Our study found that the gap between the seminiferous tubules of Hu sheep was larger than that of Tibetan sheep, and the collagen type III alpha 1 chain (*COL3A1*) was upregulated in the HS group. Therefore, we speculated that the collagen encoded by the *COL3A1* gene was the supporting framework of seminiferous tubules, which ensured spermatogenesis.

We found that the DEGs were significantly enriched in related mitochondrial regulated pathways, such as cAMP and P53. Mitochondria are highly dynamic organelles that frequently fuse and divide in different cellular processes and maintain the homeostasis of the mitochondrial network [24]. At present, it is believed that cAMP signaling in the cytoplasm can regulate dynamic changes in the mitochondria [25]. Thus, it can be speculated that the cAMP signal may affect the development of spermatogenic cells by regulating the dynamic changes of spermatogenic cell mitochondria during spermatogenesis. We also found that the *SOX9* gene was enriched in this pathway, which belongs to the Sox protein family [26]. It also controls important testicular processes, such as cell migration, retinoic acid degradation, and testicular protection from exogenous stimulation [27,28,29], and *SOX9* loss leads to male infertility [30]. Mitochondria are key organelles that regulate apoptosis. BCL2 family proteins (BAX, BAK, BCL-XS, BAD, and BOK) play an important role in the mitochondria-dependent apoptosis internal signaling pathway [31,32], and BCL2 proteins are located on the mitochondrial membrane and can inhibit apoptosis [33]. We found that the *BCL2* gene was enriched in the p53 signaling pathway and significantly upregulated in the HS group. Histological observation showed that the number of spermatogenic cells and Sertoli cells in TS was significantly lower than in HS, and *BCL2* gene expression level had a similar tendency for the number of spermatogenic cells and Sertoli cells in all samples (Figure 6). Sertoli cells provide energy and nutrients for spermatogenesis and secrete a variety of substances that participate in the differentiation and maturation of spermatogenic cells to ensure the occurrence of sperm [34]. Sertoli cells in the testis of Hu sheep provide the necessary environment for spermatocytes. Studies have found that the diameter, area, and thickness of the epithelium of seminiferous tubules are related to the number of spermatogenic cells in seminiferous tubules [35]. We found that the diameter and area of the convoluted seminiferous tubules and the thickness of the spermatogenic epithelium of HS were significantly higher than those of TS. Therefore, we speculated that the excessive apoptosis of germ cells in the testis of Tibetan sheep might be one of the important causes of affecting spermatogenesis quality.

In most mammalian tissues, histamine (HA) is a pleiotropic biogenic amine specifically synthesized by histidine decarboxylase (HDC) [36]. Testicular mast cells are a major source of HA [37]. Importantly, HA has been previously suggested to affect male fertility, such as gonadal development, spermatogenesis, and sexual behavior. Subsequent studies further revealed that HA triggers positive or negative interactions with the LH/hCG signaling pathway depending on its concentration, thereby contributing to the control of testicular androgen levels [38]. Studies have also shown that HDC is significantly expressed in the spermatids and spermatozoa of male mice. In addition, the lack of HDC resulted in lower than normal testicular interstitial cell function owing to a lack of endogenous HA [39]. In our study, HDC was not expressed in the Tibetan sheep group; therefore, we speculated that the lack of HDC might be an important reason for the mediocre spermatogenesis quality of Tibetan sheep. γ-glutamyl transferase (GGT5) plays an important role in the catabolism of glutathione, which is known to be a critical antioxidant [40]. Testicular GGT5 is expressed only at low levels in interstitial cells. GGT5 overexpression induces heme oxygenase 1 (HO-1) expression, which inhibits cytochrome P450 monooxygenase activity and severely impairs testicular steroidogenesis [41]. Testicular steroidogenesis is essential for normal spermatogenesis [42]. In this study, the *GGT5* gene was upregulated in Tibetan sheep. Therefore, the differential expression of *GGT5* may cause reproductive differences in sheep.

## 5. Conclusions

In summary, we observed changes in the histological structure within the TS and HS testicular tissues. Hu sheep had a significantly higher number of spermatogenic cells than Tibetan sheep. A total of 466 DEGs were identified in TS and HS testicular tissues. Functional analyses of these genes showed that “protein digestion and absorption”, “cAMP signaling pathway”, “focal adhesion”, and “p53 signaling pathway” were significantly enriched. We identified several DEGs, such as *COL1A1*, *COL1A2*, *COL3A1*, *SOX9*, *BCL2*, *HDC*, and *GGT5*, that were related to the spermatogenesis process, and we speculated that these genes affect the reproduction of Tibetan sheep and Hu sheep. This study provides a valuable transcriptomic reference for sheep with different fertility rates and identifies key genes and pathways that regulate spermatogenesis in sheep with different fertility rates. These identified candidate genes could be used as genetic markers for improving reproductive traits in sheep husbandry. At the molecular level, it will provide more fundamental information for improving the regulatory mechanism of spermatogenesis in Tibetan sheep testis.

## Figures and Tables

**Figure 1 genes-13-02212-f001:**
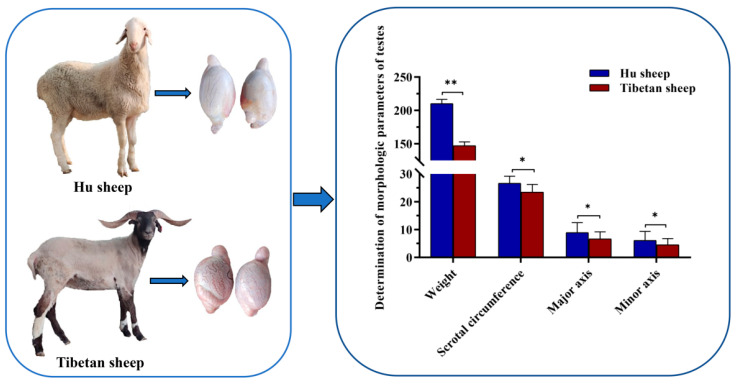
Determination of morphologic parameters of testes in Hu sheep and Tibetan sheep. The unit of weight is g, the unit of scrotal circumference is cm, and the unit of the major axis and minor axis is mm. Phenotypic values are shown as Mean ± SD. *n* = 3 sheep, * *p* < 0.05, ** *p* < 0.01.

**Figure 2 genes-13-02212-f002:**
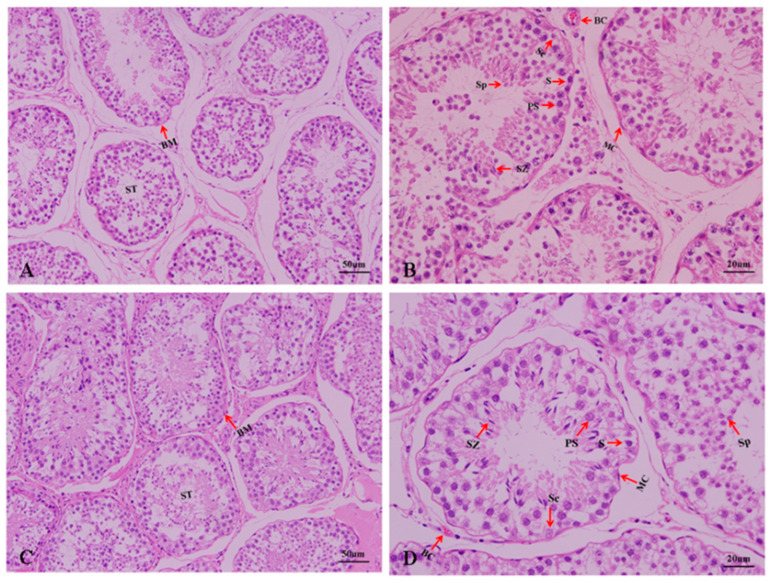
Histological observation of testicular tissue of sheep with different fecundities. (**A**,**B**): Hu sheep testes. (**C**,**D**): Tibetan sheep testes. (**A**,**C**) represent ×200 magnification. (**B**,**D**) represent ×400 magnification. ST: seminiferous tubules; BC: blood capillary; BM: basement membrane; MC: myoid cells; S: spermatogonia; PS: primary spermatocytes; SZ: spermatozoa; Sp: sperm cell; Sc: Sertoli cells.

**Figure 3 genes-13-02212-f003:**
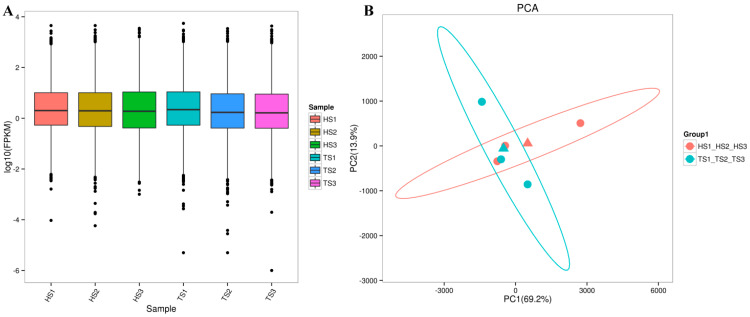
Transcriptomic analysis of the testis in sheep. (**A**) Distribution of gene expression levels. (**B**) PCA analysis of all mapped genes in HS and TS groups. Triangles represent the mean of each group of samples.

**Figure 4 genes-13-02212-f004:**
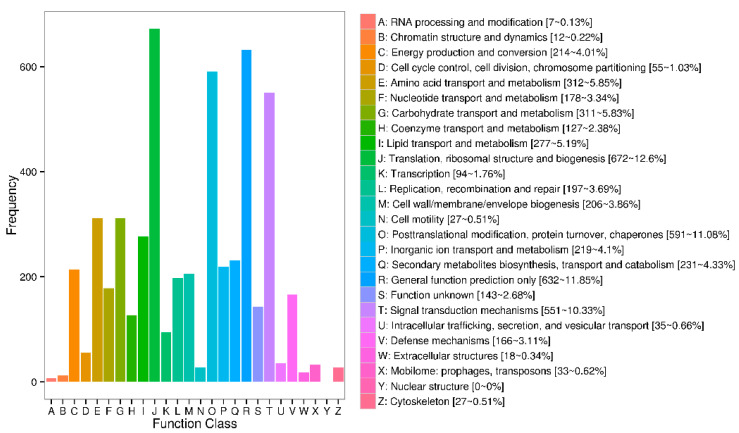
Clusters of orthologous groups (COG) function classification of transcriptome unigenes.

**Figure 5 genes-13-02212-f005:**
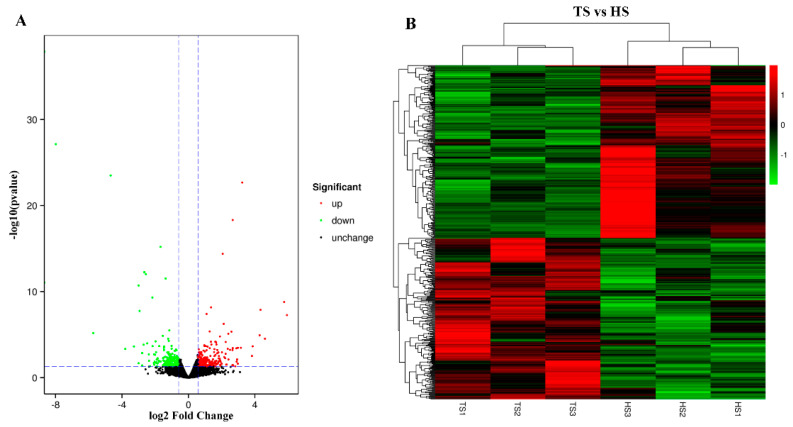
Differential expression of genes between Hu and Tibetan sheep groups. (**A**) Volcanic plot of the differentially expressed genes. (**B**) Heat map of the differentially expressed gene clustering analysis. TS: Tibetan sheep group; HS: Hu sheep group.

**Figure 6 genes-13-02212-f006:**
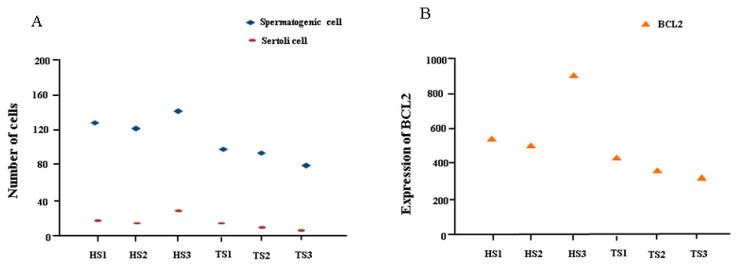
The expression levels of *BCL2* in all samples and its relationship with spermatogenic cells and Sertoli cells. (**A**) The number of spermatogenic cells and Sertoli cells in all samples. (**B**) The expression levels of *BCL2* in all samples.

**Figure 7 genes-13-02212-f007:**
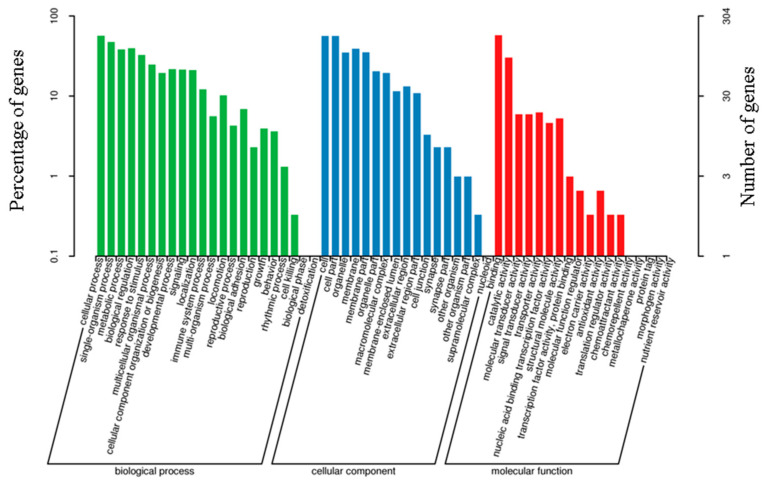
GO annotation of the differentially expressed genes.

**Figure 8 genes-13-02212-f008:**
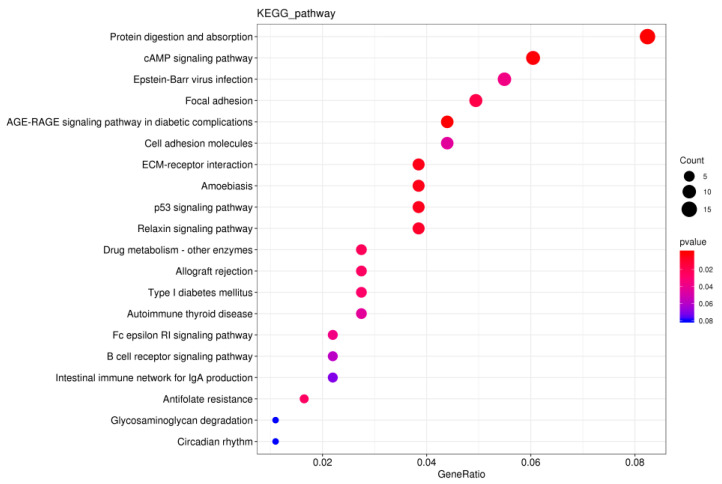
KEGG enrichment analysis of the differentially expressed genes.

**Figure 9 genes-13-02212-f009:**
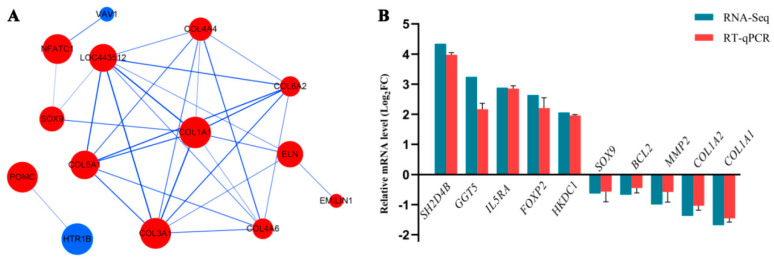
(**A**) Analysis diagram of protein–protein interaction network. Red indicates upregulated genes and blue indicates downregulated genes. (**B**) Validation of differentially expressed genes by RT-qPCR.

**Table 1 genes-13-02212-t001:** Summary information of the three Hu sheep and three Tibetan sheep used in the present study.

Breeds	Birthplace	Number	Sex	Body Weight	Age at Slaughter
Hu sheep	Minqin, Gansu	3	male	63.23 ± 2.30	360 ± 5 d
Tibetan sheep	Linxia, Gansu	3	male	46.58 ± 4.17	360 ± 5 d

**Table 2 genes-13-02212-t002:** Primer information used for RT-qPCR.

Gene Name	Primer Sequences (5′–3′)	Product Length (bp)	Annealing Temperature (°C)
*FOXP2*	F: GAGATTGCCCCAAACTACGAG R: GCAAATGTCCGTGTAAACCAG	132	54
*GGT5*	F: AGCTTCCTGCACAGCCCGTTC R: ACCTTCCTTGGCGATGTCCTCC	182	58
*HKDC1*	F: AGAAGAAATTACCTCTTGGCCTA R: CTCGCCTTAAACTTCTTGGTC	134	49
*IL5RA*	F: GTGAATTTAACTTGCACCACA R: GACATTCTTCAATCCGAGAGC	137	51
*SH2D4B*	F: CTTCCTTAACTGCAAGCCAGA R: CCATCATTGGCAGTCTACCTC	130	51
*BCL2*	F: TGAGGCTTATGAACATTCCAGT R: TTCTTCCTCCCACCCCTGCAA	121	51
*COL1A1*	F: CGCTCCTTGTGTAACTGCAT R: TTCACATGAGTCCCCATCCAC	281	57
*COL1A2*	F: GCCTATCCTTGATATTGCACCT R: CTTTTGCCCACAATTTAAGCAAG	220	53
*MMP2*	F: CATCTGGCGAACAGTGACACC R: AAGAACACAGCCTTCTCCTCC	129	57
*SOX9*	F: TGTCTAAATTCATCTGCTCCC R: TGAGCCTAAATAGACTCTGC	271	56
*β-actin*	F: TGATGATCGCAGAAAGAACCC R: CTCGCTTTGAAGGTTTCCAGT	133	52

**Table 3 genes-13-02212-t003:** Comparison of the histological parameters of testicular seminiferous tubules between Hu sheep and Tibetan sheep.

Parameters	HS	TS
Diameter of seminiferous tubules (µm)	152.26 ± 19.64 ^a^	126.35 ± 11.64 ^b^
Cross-sectional area of seminiferous Tubule (µm^2^)	16,716.71 ± 3408.55 ^a^	13,482.91 ± 1780.51 ^b^
Seminiferous epithelium thickness (µm)	38.82 ± 5.13 ^a^	34.42 ± 6.23 ^b^
Spermatogonia area (µm^2^)	13.47 ± 2.80 ^a^	11.51 ± 2.572 ^b^
Primary spermatocyte area (µm^2^)	29.27 ± 5.89 ^a^	25.59 ± 3.20 ^b^
Sperm cell diameter area (µm^2^)	9.91 ± 2.25	8.78 ± 1.52
Number of spermatogonias	37.78 ± 5.27 ^a^	32.08 ± 4.90 ^b^
Number of primary spermatocytes	26.46 ± 4.23 ^a^	22.96 ± 5.27 ^b^
Number of spermatides	30.02 ± 5.14 ^a^	24.96 ± 4.78 ^b^
Number of Sertoli cells	22.83 ± 3.72 ^a^	15.62 ± 2.73 ^b^

Note: Phenotypic values are shown as Mean ± SD. Different letters in the same row indicate significant differences (*p* < 0.05).

**Table 4 genes-13-02212-t004:** Summary of sequenced RNA-Seq data.

Items	HS1	HS2	HS3	TS1	TS2	TS3
Raw reads	43,732,762	44,297,084	44,120,746	42,299,378	45,277,764	46,281,832
Clean reads	21,866,381	22,148,542	22,060,373	21,149,689	22,638,882	23,140,916
Q20 (%)	98.42	98.16	97.48	98.29	97.41	97.38
Q30 (%)	95.09	94.71	93.22	94.89	93.08	92.96
GCcontent (%)	50.01	51.14	50.60	50.20	50.32	49.61
Mapped Reads (%)	97.45	95.97	96.02	96.93	96.02	96.02
Uniq Mapped Reads (%)	94.99	93.61	93.56	94.47	93.45	93.49
Multipl Mapped Reads (%)	2.46	2.36	2.46	2.46	2.57	2.53

## Data Availability

All the raw data obtained in this study were deposited in the NCBI Sequence Read Archive (SRA) under the Bioproject accession number PRJNA841692.

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
