# Peer review of "Transcriptomic Study of Spermatogenesis in the Testis of Hu Sheep and Tibetan Sheep"

_genes, 2022, doi:10.3390/genes13122212_

Round 1

Reviewer 1 Report

The contributions of the results section to general animal husbandry should be explained in more detail. In this context, the conclusion part of the study should be expanded and it should be clearly revealed to what extent the objectives stated at the beginning have been achieved.

Author Response

Response to Editor

My sincere gratitude goes out to you for your thoughtful work. Due to an oversight on our part, not all sources of funds for which this manuscript was funded were listed, and we have refined all of them, thank you!

Response to Reviewer 1 Comments

Reviewer #1: The contributions of the results section to general animal husbandry should be explained in more detail. In this context, the conclusion part of the study should be expanded and it should be clearly revealed to what extent the objectives stated at the beginning have been achieved.

Author Response: Thank you for your valuable and thoughtful comments. According to your helpful advice, we have added more details on the contribution of this study to the animal husbandry in the conclusion section (Lines 361-367). Thank you again!

Reviewer 2 Report

The submitted paper works on studying histology and gene expression for two species of sheep. Overall, this was a very nice study. I enjoyed reading it, and actually read it twice. I just have very minor comments below: 

1) The figure x and y axis are very small and hard to read. This can be improved. 

2) what kit for RNA and what were the quality scores. 

3) Confused by Figure 2 with the 2 y axis. Can this be made into two figures? 

Author Response

Response to Reviewer 2 Comments

The submitted paper works on studying histology and gene expression for two species of sheep. Overall, this was a very nice study. I enjoyed reading it, and actually read it twice. I just have very minor comments below: 

Author Response: Thank you very much for your comments and for recognizing our work.

  • The figure x and y axis are very small and hard to read. This can be improved. 

Author Response: Thank you for pointing this out. According to your helpful advice, we have improved these figures (Figures 3, 4, 5, 7, and 8).

  • what kit for RNA and what were the quality scores. 

Author Response: Thank you very much for your comment. Based on your suggestion, we have added information about the kit used to extract RNA and the quality scores in the new manuscript (Lines 94-95, and 97-98).

  • Confused by Figure 2 with the 2 y axis. Can this be made into two figures? 

Author Response: Thank you for pointing this out. We have made it into two figures based on your suggestions (figures 6A and B). Thank you again!
